# Glucose Metabolism Derangements and Thyroid Nodules: Does Sex Matter?

**DOI:** 10.3390/jpm12060903

**Published:** 2022-05-30

**Authors:** Alberto Gobbo, Irene Gagliardi, Andrea Gobbo, Roberta Rossi, Paola Franceschetti, Sabrina Lupo, Martina Rossi, Marta Bondanelli, Maria Rosaria Ambrosio, Maria Chiara Zatelli

**Affiliations:** 1Section of Endocrinology, Geriatrics and Internal Medicine, Department of Medical Sciences, University of Ferrara, 44100 Ferrrara, Italy; alberto.gobbo@edu.unife.it (A.G.); irene.gagliardi@edu.unife.it (I.G.); bndmrt@unife.it (M.B.); mbrmrs@unife.it (M.R.A.); 2Department of Biomedical Sciences, Humanitas University, 20090 Milan, Italy; gobbo.ndr@gmail.com; 3Department of Urology, IRCCS Humanitas Research Hospital, 20090 Milan, Italy; 4Endocrine Unit, Department of Oncology & Specialty Medicines, Azienda Ospedaliero Universitaria, 44100 Ferrara, Italy; rssrrt1@unife.it (R.R.); p.franceschetti@ospfe.it (P.F.); sabrina.lupo@unife.it (S.L.); rssmtn82@gmail.com (M.R.)

**Keywords:** impaired fasting glucose, impaired glucose tolerance, Type 2 diabetes mellitus, thyroid nodules, sex

## Abstract

(1) Background: Glucose metabolism derangements (GMD) and thyroid nodules (TNs) are the most frequent endocrine disorders, and their relationship is still controversial; little evidence is reported regarding sex differences. We aim to evaluate the association between GMDs and TNs according to sex and the sex differences in glucose metabolism and insulin sensitivity (IS). (2) Methods: We evaluated 342 patients (268 females and 74 males) at high GMD risk undergoing an oral glucose tolerance test and a thyroid ultrasound. (3) Results: The TN prevalence was 61% (*n* = 210), with no significant differences according to sex and GMD classes. The TN presence is significantly associated with age and impaired fasting glucose (IFG) in females. Males and females with normal fasting glucose (NFG) had a significantly lower OR of having TNs than females with IFG. IFG females had a significantly higher predicted probability of having TNs than NFG males and females but not IFG males. Impaired glucose tolerance/Type 2 diabetes mellitus (IGT/T2DM) is significantly associated with age and male sex, while IFG is associated with age. Females had significantly lower HOMA-index values than males. (4) Conclusions: No significant association between IGT/T2DM and TNs according to sex was found. IFG seems to play a role in TN development independently of sex. Further studies are needed to explore the relationship between TNs and GMD to identify subgroups with a higher TN risk.

## 1. Introduction

In the last years, medical research focused on gender-specific medicine, including the influence of psychological, behavioral, social and cultural characteristics, in order to pursue an individualized approach to the patient [1]. Indeed, energy metabolism is different in males as compared to females, concerning energy storage and expenditure, with a consequent strong impact on body composition [2]. Estrogens influence metabolism by improving body weight and fat tissue control, insulin sensitivity (IS) and glucose tolerance, protecting females from diet-induced obesity, liver steatosis and type 2 diabetes mellitus (T2DM) [3]. Indeed, although in 2016 women were globally more frequently obese than males [4], a report from 2017 showed a higher T2DM prevalence in males [5], supporting the hypothesis that females display higher IS than males. It is noteworthy that impaired fasting glucose (IFG) is more frequent in males, while impaired glucose tolerance (IGT) is more frequent in females [1,6]. Aging reduces differences between sexes, especially at menopause onset [7,8,9,10], and estrogen replacement therapy seems to improve the metabolic profile in post-menopausal women [8].

Thyroid disease is more frequent in females than in males: the frequency of palpable nodules reaches 5% in females vs. 1% in males in iodine-sufficient areas [11]. In addition, thyroid nodules (TNs) incidence progressively increases with age in both sexes, reaching a prevalence of >50% in ≥60-year-old subjects, as assessed by ultrasonography (US) [12,13]. Females present a higher TN prevalence than males at any age [14]. Interestingly, it has been previously reported that malignant TN incidence is higher in females (especially >40 years old) than in males [14] and that benign TNs seem to grow slower after menopause than in pre-menopausal age [15]. Estrogens might play a role in the proliferation of thyroid follicular epithelium [16,17], supporting the evidence that the TN rate is higher in females than in males. T2DM and TNs are the most frequent endocrine disorders, and several studies investigated the relationship between these two conditions [18]. Some studies, focusing on malignant TNs, reported a higher risk of thyroid cancer in patients with T2DM, but only in females and not in males [19]. Insulin-resistance (IR) plays a pivotal role in causing hyperinsulinemia, which, in turn, could stimulate thyroid follicular cell proliferation [20]. Furthermore, metabolic syndrome seems to be associated with TNs [21,22,23].

The association between glucose metabolism derangements (GMD) and TNs is still controversial and scant evidence of a relationship between these two conditions according to sex is available, thus far [24,25,26,27]. Increasing the knowledge regarding these conditions would allow identifying population subgroups at higher TN risk. Therefore, the primary aim of our study is to evaluate the possible association between GMD and TNs according to sex. As a secondary outcome, we evaluated sex differences in glucose metabolism and IS in a population at high GMD risk.

## 2. Materials and Methods

In the present study, we retrospectively reviewed data from the database of the “Azienda Ospedaliero-Universitaria, Ferrara”. We selected all patients that performed a 3-h oral glucose tolerance test (OGTT) for the presence of at least two components of the metabolic syndrome and a thyroid US because they were suspected to have TNs in our institution between 2007 and 2021. We included patients >15 years old and with normal TSH levels (0.24–4.50 µU/mL) regardless of replacement therapy. Exclusion criteria were thyroid malformations, acromegaly, recent pregnancy, central hypothyroidism, Graves’ disease or toxic multinodular goiter, antithyroid drugs therapy, previous treatment with radioactive iodine or neck radiotherapy. For each patient, we reported sex, thyroid hormone replacement therapy (THRT), age, blood glucose and insulin levels under OGTT and TN presence (absent vs. present). OGTT was performed by administering a 75 g glucose oral bolus and by drawing blood samples every 30 min for 3 h. We defined GMD according to the American Diabetes Association guidelines [28], dividing patients into three categories: normal glucose tolerance (NGT), impaired glucose tolerance (IGT) and diabetes mellitus (T2DM). IGT and T2DM patients were considered together due to numerical constraints. Fasting plasma glucose (FPG) ≥100 mg/dL identified patients with impaired fasting glucose (IFG). Patients with FPG <100 mg/dL were identified as having normal fasting glucose (NFG). To evaluate IS, HOMA index was calculated by considering FPG and fasting insulin levels; values ≥2.50 were considered as a threshold to define IR. We defined TN as any discrete lesion ≥5 mm in diameter within the thyroid gland that was radiologically distinct from the surrounding parenchyma. The US instruments used in this study were Toshiba Power Vision 6000, Hitachi Preirus^®^ (Toshiba, Roma, Italy) and Esaote MyLab Class C Advanced^®^ (Esaote, Genova, Italy).

We used the Chi-square statistic to test the distribution of categorical variables. The Mann–Whitney test and two-sample Student’s *t*-test were used to compare non-parametric and parametric variables, respectively. Non-parametric continuous variables are expressed as median value and interquartile range (IQR), while continuous variables are expressed as mean value ± standard deviation (SD). Univariable and multivariable logistic regression models were fitted for association analysis with glucose metabolism status and the presence of TN. Marginal predicted probabilities to have TN (Pr(TN)) were calculated for each interaction category based on FPG status and sex (NFG males; IFG males; NFG females; IFG females) and “contrasts for predicted margins” were used to test the differences between the four categories with IFG females as the reference group. A *p*-value <0.05 was considered statistically significant. Analyses were performed with the Excel software and STATA*17 (StataCorp, College Station, TX, Texas).

## 3. Results

In this study, we included 342 patients, 268 females (78.4%) and 74 males (21.6%), with the same prevalence of patients on THRT in both sexes. The patient median age was similar in males and females (Table 1).

### 3.1. Association between GMD and TN According to Sex

The TN prevalence was similar in males and females, and the patients with TNs showed a significantly higher median age than the patients without TNs (Table 2).

The univariable logistic regression models showed that TNs are significantly associated with age but not with sex (Table 3). When considering the OGTT categories, we found that the TN prevalence in the NGT patients was similar to the IGT/T2DM patients in both sexes (Table 2), and the univariable logistic regression model showed no correlation between the OGTT categories and the presence of TNs in our population (Table 3).

In addition, in each glucose metabolism category, males and females showed a similar TN distribution and median age. On the contrary, the IFG patients had a significantly higher TN prevalence than the NFG patients (Table 2), and the univariable logistic regression models showed a significant correlation between the IFG and TN presence in our population (Table 3).

We then divided our population into four interaction categories according to sex and FPG status (NFG males, IFG males, NFG females and IFG females). The multivariable logistic regression model adjusted for age showed that the NFG males and NFG females had a significantly lower OR of TN disease than the IFG females (Table 4).

We calculated the predicted probability of having TNs (Pr(TN)) for each group (Table 4): when analyzing the contrast between the interaction categories, the Pr(TN) in the IFG females was significantly higher than the Pr(TN) in both the NFG males and NFG females but not the IFG males (Table 5 and Figure 1).

When considering IS, the patients with IR had a TN prevalence similar to patients with normal IS (nIS) (Table 2), and the univariable logistic regression model showed no correlation between the HOMA index and TNs (see Table 3).

### 3.2. GMD and IR Distributions According to Sex

Males showed IGT/T2DM more frequently as compared to females, while there was no significant difference between males and females concerning IFG (Table 1). Age was significantly higher in the patients with GMD as compared to those without GMD (Table 1). We then divided our population into two age groups using 50 years old as the threshold value. The IGT/T2DM frequency was similar in <50-year-old males and females (34.5 vs. 20.8%, *p* = 0.12), while it was higher in males vs. females among patients ≥50 years old (66.7 vs. 43.2%, *p* < 0.01).

Both univariate and multivariate logistic regressions showed that IGT/T2DM was significantly associated with age and male sex (Table 6). Differently, age (*p* < 0.01) but not sex was significantly associated with IFG in the univariable logistic regression models.

IR was equally frequent in both sexes; the mean HOMA-index value was higher in males than in females (3.18 and 2.41, respectively). Both univariate and multivariate logistic regressions showed that male sex and IGT/T2DM are significantly associated with higher HOMA-index values. On the other hand, age did not show a significant correlation with the HOMA-index values (Table 6).

## 4. Discussion

We evaluated the association of GMD and TNs according to sex in a group of Italian patients showing risk factors for these conditions. Indeed, we retrospectively selected patients who underwent OGTT, due to the presence of at least two components of the metabolic syndrome, who also performed a thyroid US, because they were suspected to have a TN. Furthermore, we evaluated sex differences in glucose metabolism and IS.

We found a significant association between TNs and IFG in female patients but not in males. This finding is likely due to the fact that the number of IFG males in our sample is insufficient to perform a reliable statistical evaluation. Differently, we did not find any association between TNs and IGT/T2DM according to sex. Our results, therefore, would suggest that sex does not influence TN distribution in patients having different glucose metabolic statuses. On the other hand, the higher incidence of TNs in females but not in males with IFG may suggest that sex hormones could play a role in IFG patients. Indeed, it has been previously reported that in vitro androgens reduce thyroid cell proliferation, without affecting follicular cell function [29], while estrogens stimulate follicular cell proliferation [16,17]. IFG patients may therefore display a higher sensitivity to sex hormone influences on TN development than NFG patients. On the other hand, we found a different distribution according to sex only in IFG patients, who displayed a higher TN rate as compared to NFG. These results might suggest that factors associated with IFG may facilitate TN development in females. Further studies are needed to elucidate these issues.

Previous studies explored the association between GMD and TNs [20,21,22,23], but only a few of them investigated a possible sex-based association between GMD and TNs [24,25,26,27]. In the study by Ding et al. [24], T2DM and metabolic syndrome were independently associated with TNs in females but not in males, supporting the hypothesis that sex may be an important factor influencing TN development in the general population living in a rural Chinese region. On the contrary, a systematic review [25] considering 13 studies mainly conducted in Eastern Asia found a correlation between metabolic syndrome and TNs independently of sex, suggesting that sex may not significantly impact on TN development in patients with metabolic syndrome. Similarly, Buscemi et al. [26] found that T2DM and TNs were significantly associated independently of sex in an Italian patients’ group. Our results are in line with these latter findings because our patients were selected based on the presence of metabolic syndrome components and share a common genetic background. On the other hand, our study did not find significant differences in TN prevalence between NGT and IGT/T2DM patients, suggesting that TN prevalence does not associate with dynamic glucose metabolism alterations, differently from previous reports [18,26]. These differences may be due to the different study settings; indeed, we selected patients with risk factors for GMD, while the other studies did not. Zhang et al. [27] showed that metabolic syndrome (and hyperglycemia, in particular) increase the risk of TNs in both genders, with a higher Odds Ratio in females. Our results are in agreement with the reported evidence that hyperglycemia is associated with TNs. However, in our study, when considering hyperglycemic patients, we could not observe a significantly different Odds Ratio between IFG males and IFG females due to the limited sample size. Therefore, we are unable to confirm the finding that hyperglycemic females have a higher probability of having TNs then hyperglycemic males. This difference may also be explained by the fact that the study by Zhang et al. takes into consideration mainly Asian subjects, while in our study, all patients were Caucasian.

Furthermore, in our sample, we did not find any association between the HOMA index and TN presence. These results do not support the hypothesis that hyperinsulinemia and IR promote thyrocyte proliferation and TN development [20,30]. The latter has been hypothesized to also be caused by higher TSH levels associated with IR in patients with GMD [31] or by a relative iodine deficiency associated with obesity [26]. Our patients were selected based on TSH levels in the normal range, but we did not normalize for iodine intake. Therefore, we cannot exclude that iodine deficiency may influence the outcome of our study despite iodine prophylaxis being highly active in Italy [32].

We found that IFG is significantly associated with TN presence in females, which is in line with previous studies showing that metabolic syndrome and IFG are significantly associated with TN development [24,25,30,33,34]. Older age may affect TN prevalence [12,13], but our results are independent of patients’ age because they were obtained after adjusting for this parameter.

Previous findings show that GMD are more frequent in males as compared to females [3,5] and that this difference fades after menopause because females lose the natural protection provided by estrogens [7,8,9,10]. Therefore, the difference between sexes concerning GMD is expected to decrease with age. However, in our sample, the male sex was associated with IGT/T2DM even after adjusting for age and the IGT/T2DM frequency was similar between males and females in subjects <50 years old, while it was higher in male patients ≥50 years old. These results are in contrast with the expectations and suggest that the impact of aging on glucose metabolism is stronger in males as compared to females in a high-risk population like the one we enrolled. This hypothesis is in keeping with the evidence that males show a reduced beta-cell functional reserve as compared to females, independently of age [35,36]. Compensatory mechanisms might therefore exhaust more rapidly, with an anticipated onset of hyperglycemia 2 h after glucose load in males than in females.

On the contrary, IFG and IR (expressed by the HOMA index) were equally frequent in males and females, independently of age. This finding is in contrast with previous studies reporting that the male sex is associated with IFG and IR [1,3,6,35]. The similar distribution of IFG and IR between males and females may be explained by the fact that our patients were selected from subjects displaying risk factors for metabolic disorders. On the same line, the TN distribution was similar in males and females, in contrast with previous studies showing that females display a higher incidence of TNs as compared to males, independently of age [11,14]. This discrepancy may be due to different study settings. Indeed, our study selected patients undergoing US for suspected TNs, while the above-quoted studies evaluated the general population.

## 5. Conclusions

In conclusion, we did not find any association between TNs and IGT/T2DM, neither in males nor in females. Therefore, our results do not support performing TN screening in patients with dynamic GMD belonging to a population showing risk factors for both disorders.

On the other hand, we found an association between TNs and IFG females but not IFG males, most likely due to the numerical shortage of the latter group. For this reason, we cannot conclude in favor of the presence of sex differences in our patients’ group, even though the association between TNs and IFG is likely in both sexes.

Further research should be conducted in this direction. Based on our results and the available evidence, a TN screening program may be useful in patients with IFG and other risk factors for metabolic disorders, independently of sex. Future research should also focus on the probability of finding malignant TNs in patients with metabolic syndrome or its components to evaluate the cost–benefit ratio of this type of screening.

Finally, we confirm that males are more likely to be affected by dynamic GMD and, in a population with risk factors for these disorders, aging does not reduce the different GMD rates in males and females.

It is important to underline that the population included in this study is not representative of the general Italian population. Indeed, we recruited patients undergoing OGTT and thyroid US and therefore already having risk factors for GMD and suspected TNs. This allowed us to study a highly selected population, reproducing the patients group usually referred to a specialized endocrinology clinic. Consequently, our results may exclusively apply to this patient subgroup and should not be extended to the general population.

## Figures and Tables

**Figure 1 jpm-12-00903-f001:**
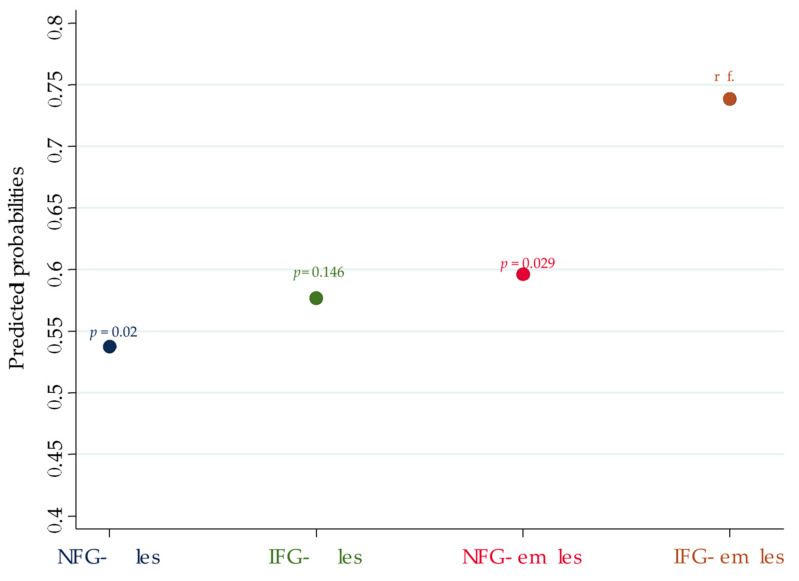
Pr(TN) of each interaction category according to sex and FPG. p-values refer to the analysis of the contrast between the Pr(TN) of the correspondent interaction category vs. the Pr(TN) of the reference group (IFG females).

**Table 1 jpm-12-00903-t001:** Distributions of age, THRT, OGTT categories, FPG categories, insulin resistance and HOMA index in males vs. females. Percentages refer to the population of the corresponding sex. Mann–Whitney test was used to analyze age of each nominal group.

			Sex		*p*-Value	Median Age (IQR)	*p*-Value
			Males *n* = 74	Females *n* = 268			
Median Age (IQR)		TOT *n* = 342	52 (44.5–62.25)	51 (36–61)	n.s.	–	
THRT	NO	254	55 (74.3%)	199 (74.3%)	n.s.	51.5 (37–62)	n.s
	YES	88	19 (25.7%)	69 (25.7%)		51 (40.25–60.5)	
OGTT	NGT	213	34 (45.9%)	179 (66.8%)	<0.01	49 (32–57.5)	<0.01
	IGT/T2DM	129	40 (54.1%)	89 (33.2%)		56 (48.5–65.5)	
IFG	NO	247	50 (67.6%)	197 (73.5%)	n.s.	49 (33–59)	<0.01
	YES	95	24 (32.4%)	71 (26.5%)		56 (50–63)	
Insulin sensitivity (HOMA index)	IS	208	45 (60.8%)	163 (60.8%)	n.s.	51 (39.25–61.75)	n.s.
	IR	134	29 (39.2%)	105 (39.2%)		53 (36–62)	
Median HOMA index (IQR)			2.06 (1.54–4.25)	1.98 (1.34–3.25)	n.s.	–	

**Table 2 jpm-12-00903-t002:** TN distribution according to sex, OGTT categories, FPG categories and HOMA index. Percentages refer to the nominal group in the corresponding line.

			TN		*p*-Value
			NO	YES	
Median Age (IQR)		TOT	43.5 (25.5–56)	54 (46–62)	<0.001
Sex	Males	74	32 (43.2%)	42 (56.8%)	n.s.
	Females	268	100 (37.3%)	168 (62.7%)	
OGTT	NGT	213	85 (39.9%)	128 (60.1%)	n.s.
	IGT/T2DM	129	47 (36.4%)	82 (63.6%)	
OGTT in males	NGT	34	13 (38.2%)	21 (61.8%)	n.s.
	IGT/T2DM	40	19 (47.5%)	21 (52.5%)	
OGTT in females	NGT	179	72 (40.2%)	107 (59.8%)	n.s.
	IGT/T2DM	89	28 (31.5%)	61 (68.5%)	
IFG	NO	247	108 (43.7%)	139 (56.3%)	0.002
	YES	95	24 (25.3%)	71 (74.7%)	
HOMA index	IS	208	85 (40.9%)	123 (59.1%)	n.s.
	IR	134	47 (35.1%)	87 (64.9%)	

**Table 3 jpm-12-00903-t003:** Univariable logistic regression models (ULRM) with TNs as the main outcome.

Outcome Variable	Independent Variable	ULRM		
		OR	*p*-Value	C.I.
TN	Females	1.28	0.354	0.759–2.158
	Age	1.039	<0.001	1.024–1.054
	IGT/T2DM	1.159	0.523	0.738–1.82
	IFG	2.299	0.002	1.357–3.892
	HOMA index	1.017	0.764	0.910–1.137

**Table 4 jpm-12-00903-t004:** Multivariable logistic regression model (MLRM) with TN as the dependent variable. Age and the four interaction categories according to sex and FPG status are the independent variables. IFG females were used as the base reference. Predicted probabilities of having TN (Pr(TN)) were calculated for each interaction category.

Dependent Variable	Independent Variable	MLRM			Predictive Margins	
		OR	*p*-Value	C.I.	Pr(TN)	C.I.
TN	NFG males	0.385	0.022	0.17–0.872	0.538	0.405–0.67
	IFG males	0.458	0.134	0.164–1.273	0.577	0.384–0.769
	NFG females	0.499	0.04	0.257–0.968	0.596	0.529–0.663
	IFG females	reference	//	//	0.739	0.631–0.846
	Age	1.035	<0.01	1.02–1.051	//	//

**Table 5 jpm-12-00903-t005:** Analysis of the contrast between the predicted probability of having TN (Pr(TN)) in IFG females vs. the other interaction categories.

	Contrast of Predictive Margins		
	Contrast	*p*−Value	C.I.
NFG males vs. IFG females	−0.201	0.02	−0.371; −0.031
IFG males vs. IFG females	−0.162	0.146	−0.379; +0.056
NFG females vs. IFG females	−0.142	0.029	−0.271; −0.014

**Table 6 jpm-12-00903-t006:** Univariable and multivariable logistic regression models with IGT/T2DM, IFG and HOMA index as main outcomes.

–	Independent Variable	ULRM			MLRM		
		OR or Coefficient *	*p*-Value	C.I.	OR or Coefficient *	*p*-Value	C.I.
IGT/T2DM	Female sex	0.423	0.001	0.25–0.71	0.462	0.008	0.261–0.82
	Age	1.043	<0.001	1.027–1.059	1.039	<0.001	1.022–1.058
	IFG	3.886	<0.001	2.37–6.38	2.696	<0.001	1.588–4.579
	HOMA index	1.264	<0.001	1.11–1.438	1.239	0.003	1.075–1.429
HOMA index	Female sex	−0.761 *	0.003	−1.269; −0.253	−0.605 *	0.02	−1.113; −0.097
	Age	+0.001 *	0.849	−0.012; +0.014	−0.007 *	0.290	−0.021; +0.006
	IGT/T2DM	+0.861 *	<0.001	+0.434; +1.288	+0.841 *	<0.001	+0.390; +1.291

Numbers with * correspond to Coefficients, those without correspond to OR.

## Data Availability

Data that support the findings of this study are available on request from the corresponding author.

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
