# Peer review of "Glucose Metabolism Derangements and Thyroid Nodules: Does Sex Matter?"

_jpm, 2022, doi:10.3390/jpm12060903_

Round 1

Reviewer 1 Report

The authors describe a retrospectively reviewed data from the database at the “Azienda Ospedaliero-Universitaria, Ferrara”. The authors found a significant association between TN and IFG in female patients, but not in males. However, they explain that this might be due to the fact that the number of IFG males in their sample is insufficient to perform a reliable statistical evaluation.

Major concerns

The novelty is a concern. I encourage the authors to remark what is novel compared to previously publish data. As one example, I suggest to highlight what similarities/discrepancies compared to the published article Front. Endocrinol., 21 September 2021 | https://doi.org/10.3389/fendo.2021.736972

Please include in the abstract the number of males and females utilized in the study. 

Minor concerns

Abbreviations should be minimized to make the manuscript less difficult to read.

This manuscript must be proof read by a native English speaker. 

Author Response

Answers to Reviewer 1

We thank the Reviewer for accurately reading our manuscript and providing useful remarks. We hereby provide a point-by-point reply.

Major concerns

  • The Reviewer states that the novelty of our manuscript is a concern and encourages to compare our findings with those shown in Front. Endocrinol., 21 September 2021 https://doi.org/10.3389/fendo.2021.736972

We thank the Reviewer for bringing to our attention this very interesting study, showing that subjects with Metabolic Syndrome (MetS) have a higher prevalence of TNs vs. subjects without MetS, taking into account gender disparities. We added this manuscript to the list of references, which changed accordingly. Our results are in agreement with the reported evidence that hyperglycemia is associated with TNs. However, in our study, when considering hyperglycemic patients, we could not observe a significantly different Odds Ratio between males and females, due to the limited sample size. Therefore, we are unable to confirm the finding that hyperglycemic females have a higher probability of having TN as compared to hyperglycemic males. This difference may also be due to the fact that the study by Zhang et al. takes into consideration mainly Asian subjects, while in our study all patient were Caucasian.

We do not have data concerning the other components of MetS, therefore we cannot compare further our results. We added these considerations to the revised manuscript.

  • The reviewer requires that we include in the abstract the number of males and females enrolled in the study. 

Following the reviewer’s indication, we added the number of enrolled males and females in the abstract.

Minor concerns

  • The Reviewer suggests minimizing abbreviations

As indicated in the reply to the Editor’s comments, we removed many abbreviations and used the whole definition in the revised manuscript.

  • The Reviewer suggests improving English language.

The entire manuscript has been reviewed and imprecisions have been amended.

Reviewer 2 Report

Thyroid nodules and prediabetes/diabetes together with obesity are the most common endocrine disorders. While they might be possibly related in some subjects, the exact mechanisms behind such associations are elusive.

The authors of the manuscript try and fail to elucidate such a relationship on clinical basis – a fair result by itself. My main concern about the study is the participant selection.  The studied cohort is heavily biased, as sincerely pointed out by the authors – the prevalence of diabetes and prediabetes and thyroid nodules is much higher than in the general population. That means that they have been preselected according to higher probability of having these disorders. Therefore, the results can hardly be extrapolated to even a presumably similar cohort.

A similar but much larger, more complex and better designed study was published by Chang X et al. (ref 33).

Materials

Please define the TN. "Suspected TN" was mentioned on Line 76. Did you set a lower size margin above which nodules were registered (e.g. 5 mm)?

A large proportion of the participants received THRT. Was Hashimoto's thyroiditis the cause? How was nodularity on the background of Hashimoto determined?

Line 94. Student’s t-test. Please correct. Moreover, TNs are more prevalent in women than in men even in the absence of IFG.

Line 96. Continuous variables. Please, remove “parametric”.

What proportion of the participants were overweight or obese?

Results

The distribution of HOMAi looks skewed. Please check and if confirmed, present as a media and IQR.

Line 151. It is true that the medians of the age in both genders were close to 50. Nevertheless, you either use the median or some other unrelated threshold.

Discussion

Lines 179-184.  It might be tempting to conclude that IFG or NGT interact in some way worth the effect of sex steroids but the cohort selection precludes that.

Please, remove ref. 12 – it does not support the statement since the paper discusses the palpable nodule in the dawn of US neck exams.

Ref. 36 does not support the statement in the text.

Please check the tenses in the text.

Author Response

Answers to Reviewer 2

We thank the Reviewer for the accurate revision of our manuscript. We herby provide a poin-by point reply to his/her useful comments.

  • Thyroid nodules and prediabetes/diabetes together with obesity are the most common endocrine disorders. While they might be possibly related in some subjects, the exact mechanisms behind such associations are elusive. The authors of the manuscript try and fail to elucidate such a relationship on clinical basis – a fair result by itself. My main concern about the study is the participant selection.  The studied cohort is heavily biased, as sincerely pointed out by the authors – the prevalence of diabetes and prediabetes and thyroid nodules is much higher than in the general population. That means that they have been preselected according to higher probability of having these disorders. Therefore, the results can hardly be extrapolated to even a presumably similar cohort. A similar but much larger, more complex and better designed study was published by Chang X et al. (ref 33).

We thank the Reviewer for this remark, which indeed underlines the limitations of our study. We agree with the Reviewer that our sample is heavily biased, as recognized also in the submitted manuscript. And indeed, our study population represents a sample of individuals referring to our Endocrine center due to a health problem (i.e. thyroid nodule and/or hyperglycemia/diabetes), therefore any retrospective study (like this one) at our center will suffer from this bias. We agree that our results cannot be extrapolated to the general population. However, we believe that our study population may reflect the patient group composition referring to an Endocrine center in the daily practice. The novelty of the study is the suggestion that investigating for TN presence those patients referring for glucose derangements could be useful in patients with IFG, independently of sex.

The study by Chang et al. is indeed designed differently and with a much greater population, therefore provides different results. Our study has a retrospective design and cannot be compared to the study by Chang et al. However, we would like to underline that they found similar results, since this study shows that the detection rate of TN in the PreDM population was significantly higher than that of NGT.

  • Please define the TN. "Suspected TN" was mentioned on Line 76. Did you set a lower size margin above which nodules were registered (e.g., 5 mm)? A large proportion of the participants received THRT. Was Hashimoto's thyroiditis the cause? How was nodularity on the background of Hashimoto determined?

In keeping with the Reviewer’s comment, we added the definition of TN according to the ATA guidelines. In addition, we considered as nodules only those with a size ≥5 mm. The mention of “suspected thyroid nodule” relates to the indication to perform a thyroid ultrasound and does not relate to nodule diameter. We modified the phrase to clarify this issue (patients suspected to have a TN).

Approximately one fourth of the enrolled patients was on THRT due to primary hypothyroidism, in the large majority on the background of Hashimoto’s thyroiditis. In this setting, nodularity was determined following the above quoted guidelines. Only 12 patients with TN on THRT were not submitted to FNA, therefore the nature of the nodules assessed in the other patients was defined by cytology and excluded the presence of thyroiditis-related nodular images.

  • Line 94. Student’s t-test. Please correct. Moreover, TNs are more prevalent in women than in men even in the absence of IFG.

We thank the reviewer for spotting this oversight, which has been corrected in the revised manuscript. Concerning TN prevalence, in our sample we did not find a significant difference between males and females in the general population as well as in patients with normal fasting glucose.

  • Line 96. Continuous variables. Please, remove “parametric”.

In keeping with the Reviewer’s indication, we removed “parametric”.

  • What proportion of the participants were overweight or obese?

We agree with the reviewer that it would be interesting to evaluate also patient’s weight, as well as other MetS parameters. However, due to the retrospective nature of our analysis, this parameter was not available for all the patients and therefore we decided not to include it in the analysis.

  • The distribution of HOMAi looks skewed. Please check and if confirmed, present as a median and IQR.

In keeping with the Reviewer’s comment, we checked again the distribution of HOMAi and confirmed that it is asymmetric. Therefore, following the Reviewer indication, in the revised Table 1 we present HOMAi as median and IQR in the revised manuscript. Consequently, we analyzed the data by the Mann Whitney test, which failed to assess any significant difference between males and females. However, when we applied the univariable regression logistic model we found that sex is significantly associated to HOMAi, as already pointed out in the submitted manuscript.

  • Line 151. It is true that the medians of the age in both genders were close to 50. Nevertheless, you either use the median or some other unrelated threshold.

We thank the Reviewer for this remark. In keeping with his/her suggestion, we removed the reference to the median age of our sample population, and simply referred to 50 years as an arbitrary threshold.

  • Lines 179-184.  It might be tempting to conclude that IFG or NGT interact in some way worth the effect of sex steroids, but the cohort selection precludes that.

We thank the Reviewer for this remark. Indeed, our results suggest that IFG may influence the likelihood of finding a TN. As pointed above, the cohort selection reflects the composition of the patient group presenting to an endocrine center. Therefore, our results cannot thoroughly identify a possible causative link between glucose metabolism derangements and the occurrence of TN. In addition, we clearly state that our results would suggest that sex does not influence TN distribution in patients having a different glucose metabolic status (line 181 of the submitted manuscript).

  • Please, remove ref. 12 – it does not support the statement since the paper discusses the palpable nodule in the dawn of US neck exams.

In keeping with the Reviewer’s indication, we removed the old ref.12 that was replaced with a more appropriate reference.

  • Ref 36 does not support the statement in the text.

In keeping with the Reviewer’s indication, we removed this reference. Reference 36 in the revised manuscript is related to a different publication.

  • Please check the tenses in the text.

We thank the Reviewer for this remark. The entire manuscript has been reviewed and imprecisions have been amended.

Round 2

Reviewer 1 Report

The have responded to previous comments and the manuscript has been improved